# Prognostic Factors in Children and Adolescents with Lymphomas and Vertical Transmission of HIV in Rio de Janeiro, Brazil: A Multicentric Hospital-Based Survival Analysis Study

**DOI:** 10.3390/cancers15082292

**Published:** 2023-04-14

**Authors:** Nathalia Lopez Duarte, Ana Paula Silva Bueno, Bárbara Sarni Sanches, Gabriella Alves Ramos, Julia Maria Bispo dos Santos, Henrique Floriano Hess e Silva, Janaina de Oliveira Pondé, José Gilberto de Sá, Priscila Mazucanti Rossi, Patricia Regina Cavalcanti Barbosa Horn, Denise Cardoso das Neves Sztajnbok, Norma de Paula Motta Rubini, Elaine Sobral da Costa, Cristiane Bedran Milito, Thalita Fernandes de Abreu, Marcelo Gerardin Poirot Land

**Affiliations:** 1Faculty of Medicine (FM), Federal University of Rio de Janeiro (UFRJ), Rio de Janeiro 21941-617, Brazil; 2Internal Medicine Postgraduate Program, Faculty of Medicine (FM), Federal University of Rio de Janeiro (UFRJ), Rio de Janeiro 21941-617, Brazil; 3Transdisciplinary Center for Research in Child and Adolescent Health (NTISCA), Institute of Pediatrics and Childcare Martagão Gesteira (IPPMG), Federal University of Rio de Janeiro (UFRJ), Rio de Janeiro 21941-612, Brazil; 4Pediatric Hematology Service, Institute of Pediatrics and Childcare Martagão Gesteira (IPPMG), Federal University of Rio de Janeiro (UFRJ), Rio de Janeiro 21941-612, Brazil; 5Faculty of Medical Sciences (FCM), Pedro Ernesto University Hospital (HUPE), State University of Rio de Janeiro (UERJ), Rio de Janeiro 20551-030, Brazil; 6Department of Infectious and Parasitic Diseases, Hospital Municipal Jesus (HMJ), Municipal Health Secretariat (SMS-RJ), Rio de Janeiro 20550-200, Brazil; 7Department of Hematology, Pedro Ernesto University Hospital (HUPE), State University of Rio de Janeiro (UERJ), Rio de Janeiro 20551-030, Brazil; 8Pediatric Infectious Diseases Division, Department of Pediatrics, Pedro Ernesto University Hospital (HUPE), State University of Rio de Janeiro (UERJ), Rio de Janeiro 20551-030, Brazil; 9Department of Allergy and Immunology, School of Medicine and Surgery, Gaffrée and Guinle University Hospital (HUGG), Federal University of the State of Rio de Janeiro (UNIRIO), Rio de Janeiro 20270-004, Brazil; 10Department of Pathology, Faculty of Medicine (FM), Clementino Fraga Filho University Hospital (HUCFF), Federal University of Rio de Janeiro (UFRJ), Rio de Janeiro 21941-617, Brazil; 11Infectious and Parasitic Diseases Service, Institute of Pediatrics and Childcare Martagão Gesteira (IPPMG), Federal University of Rio de Janeiro (UFRJ), Rio de Janeiro 21941-612, Brazil

**Keywords:** HIV, vertical transmission (VT), combined antiretroviral therapy (cART), pediatric, lymphoma, survival, prognosis, outcome, Brazil

## Abstract

**Simple Summary:**

Lymphomas related to HIV are generally aggressive and have a poor prognosis, despite the use of effective combined antiretroviral therapy (cART) and effective chemotherapy treatment. This study aimed to evaluate, for the first time, survival and prognostic factors in children and adolescents living with HIV (CLWH) with lymphomas in Rio de Janeiro (RJ), Brazil. Our study population showed reduced overall, event-free, and disease-free survival probabilities. Performance status 4 (PS 4) and low CD4+ T-cell counts were considered poor prognostic factors. This study was designed to improve the quality of HIV-related lymphoma treatment in our population.

**Abstract:**

Lymphomas related to HIV are generally aggressive and have a poor prognosis, despite the use of combined antiretroviral therapy (cART) and effective chemotherapy treatment. To determine survival and prognostic factors in children and adolescents living with HIV (CLWH) in Rio de Janeiro (RJ), Brazil, who developed lymphomas, we performed a retrospective and observational study of vertically infected CLWH aged from 0 to 20 incomplete years during1995 to 2018 at five reference centers for cancer and HIV/AIDS treatment. Of the 25 lymphomas, 19 were AIDS-defining malignancies (ADM) and 6 were non-AIDS-defining malignancies (NADM). The 5-year overall survival (OS) and 5-year event-free survival (EFS) probabilities were both 32.00% (95% CI = 13.72–50.23%), and the 5-year disease-free survival (DFS) probability was 53.30% (95% CI = 28.02–78.58%). In the multivariate Cox regression analysis, performance status 4 (PS 4) was considered a poor prognostic factor for OS (HR 4.85, 95% CI = 1.81–12.97, *p* = 0.002) and EFS (HR 4.95, 95% CI = 1.84–13.34, *p* = 0.002). For the DFS, higher CD4+ T-cell counts were considered a better prognostic factor (HR 0.86, 95% CI = 0.76–0.97, *p* = 0.017) in the multivariate Cox regression analysis. This study demonstrates, for the first time, survival and prognostic factors for CLWH who developed lymphomas in RJ, Brazil.

## 1. Introduction

When occurring concomitantly in children with immature immune systems, lymphomas and HIV can have serious consequences. HIV-related lymphomas are generally associated with immune dysregulation since HIV infection results in losses of both cellular and humoral immunity [1,2,3,4,5,6,7,8,9]. Lymphomas, most of which are of the B cell non-Hodgkin subtype in children and adolescents living with HIV (CLWH), are generally diagnosed in an advanced stage with extranodal involvement and usually have an aggressive clinical course [10,11,12,13].

In line with this, it is well-known that concurrent HIV infection significantly worsens the prognosis of children and adolescents with lymphoma in spite of antiretroviral treatment and chemotherapy [14,15,16,17,18]. Older children generally have higher death rates, mostly due to advanced-stage disease, poor performance status (PS), and lower CD4+ T-cell counts at lymphoma diagnosis [14,15,19,20]. Moreover, late presentation, treatment-related toxicity, and drug interactions may prevent childhood HIV-associated lymphomas from receiving more effective therapy, leading to increased morbidity and mortality [21].

HIV-associated cancer incidence and survival among CLWH have changed over time with the advent of increased access to and uptake of combined antiretroviral therapy (cART) [2,13,17,18,20,22,23]. Accurate antiretroviral implementation decreased morbidity and mortality by inhibiting viral replication, restoring immunological surveillance, and, consequentially, increasing life expectancy [20,24,25,26]. Thus, initiating cART as soon as possible has also been related to better recovery of CD4+ T-cell counts [20,27].

The high HIV care coverage provided by the Brazilian Ministry of Health and access to antiretroviral therapy are a great global example [28,29,30]. They have resulted in increased lifespans for CLWH, as well as possibly causing the consequent development of a variety of diseases, including some malignancies, such as lymphomas [20,24,25]. Regarding lymphoma treatment in Brazil, particularly in past decades, patients have been treated with a variety of chemotherapy protocols, with or without radiotherapy, in accordance with institutional preference.

There is a paucity of cancer survival studies on CLWH globally. Limited studies report an increased mortality risk for patients with both HIV and various cancers due to infection complications [9,31], but there are no published data on pediatric HIV and lymphomas in Brazil. The clinical presentation of CLWH with lymphoma and their outcomes are, therefore, of great interest, as they could highlight how the actual management of these children and adolescents influences disease outcome, especially in an endemic setting, as well as improve children’s quality of life and life expectancy [9,13]. The purpose of this study was to assess survival and prognosis in children and adolescents vertically infected with HIV who developed lymphomas.

## 2. Materials and Methods

### 2.1. Study Design, Study Populations, and Data

This was a multicentric hospital-based and observational study of a retrospective cohort of pediatric patients. Children and adolescents aged 1 month to 19 years, 11 months, and 29 days with HIV infection through vertical transmission (VT) and lymphomas were selected in accordance with the criteria of the Brazilian Ministry of Health/CDC [32]. These patients were all assisted at one of the five participating institutions from 1 January 1995 to 1 January 2018: Instituto de Puericultura e Pediatria Martagão Gesteira (IPPMG/UFRJ), Hospital Universitário Clementino Fraga Filho (HUCFF/UFRJ), Hospital Universitário Pedro Ernesto (HUPE/UERJ), Hospital Universitário Gaffrée e Guinle (HUGG/UNIRIO), and Hospital Municipal Jesus (HMJ/SMS-RJ). All patients included in this study started medical assistance in this period.

We collected data from medical records in order to obtain information on medical evolution and complementary exams results. These also included data on the diagnosis of HIV infection and lymphoma, histopathologic evaluation, and clinical course after the diagnosis of HIV and lymphoma.

Finally, ethical approval was obtained from each of the institutional review boards of the five centers.

### 2.2. Study Period Definition

The study period was from 1995 to 2018. During this period, cART was administrated in accordance with the widespread use of the therapy in CLWH in Brazil and around the world. Patients were classified into different eras by the date of entrance at the study institutions: from 1995 to 1999 (1995–1999) for early-period cART (Early-cART era), when the first protease inhibitors were adopted; from 2000 to 2003 (2000–2003) for mid-period cART (Mid-cART era), when cART was simplified and the non-nucleoside reverse transcriptase inhibitor (NNRTI) regimens were initiated; and, finally, from 2004 to 2018 (2004–2018) for late-period cART (Late-cART era), when new cART regimens based on coformulations of antiretroviral drugs were introduced [2,13,20,25]. We included the variable “cART era” in the prognostic analysis since treatment with cART would have prognostic implications for lymphoma survival analysis.

### 2.3. Tumor Samples and Laboratorial Analysis

Tumor samples were reanalyzed and reclassified according to the 2022 WHO classification [33] at the Department of Pathology of the HUCFF/UFRJ. We used the more recent morphological and immunohistochemical techniques for this purpose whenever possible.

### 2.4. Outcome Variables and Baseline Characteristic

We classified the cases using the ICD-11 MMS code for neoplasms of hematopoietic or lymphoid tissues. AIDS-defining malignancies (ADMs) were codified as follows: 2A85.6 for Burkitt lymphoma (BL) (13 cases) and 2A81.Z for diffuse large B-cell lymphoma (DLBCL) (6 cases). For non-AIDS-defining malignancies (NADMs), we used the following codes: 2B30.10 for nodular sclerosis classical Hodgkin lymphoma (NSCHL) (four cases), 2A90.A for anaplastic large cell lymphoma (ALCL CD30+/ALK+) (one case), and 2A90.C for peripheral T-cell lymphoma (PTCL) (one case) [10,12,34].

The immunological status was classified using the CDC clinical category and immunosuppression stage (N/A/B/C and 1/2/3/unknown, respectively) at the date of diagnosis of HIV and lymphomas. CD4+ T-cells counts were described as percentages. CD4+ T-cell percentages were those available during the clinical follow-up period. Percentages above 25% were classified as absence of immunosuppression (stage one); between 15% and 25% as moderate immunodeficiency (stage two); and below 15% as severe immunodeficiency (stage three) [32]. Finally, CD4+ values were included in the analyses with a margin of 90 days (before or after) from the date of diagnosis of HIV or lymphoma. HIV viral loads were measured using polymerase chain reaction (PCR) and are given in copies/mL [32]; they were also considered with a margin of 90 days (before or after) from the date of diagnosis of HIV or lymphoma.

Data included biological sex, year of lymphoma diagnosis, age at lymphoma diagnosis, ART/cART use, cART era, CD4+ T-cell count, HIV viral load, and CDC category/stage (at HIV and lymphoma diagnosis). Other variables included histological subtype, site of involvement at presentation, PS, stage, presence of B symptoms, lymphoma treatment features, outcomes, status at the moment of death, and others. Baseline characteristics were summarized using descriptive statistics.

### 2.5. Staging

Patients were staged according to the Murphy or Ann Arbor classification system, depending on the lymphoma subtype [35,36].

### 2.6. Treatment

Antiretroviral therapy (ART—mono- or dual-drug therapy) was initially supplied in the early 1990s by the Brazilian Ministry of Health. The cART regimen, which includes antiretroviral drugs from at least three separate classes (protease inhibitors, nucleoside reverse transcriptase inhibitors, non-nucleoside reverse transcriptase inhibitors, and integrase inhibitors) or selected dual-drug-based regimens, has been available since 1996. It is important to point out that Brazil was the first developing country to provide free and universal access to cART [28,29,30].

In this study, antiretroviral therapy and comprehensive HIV care and treatment were provided following the respective Brazilian Ministry of Health guidelines according to each period [32].

The chemotherapy protocols used in this study included NHL-BFM 90, NHL-BFM 95, B-NHL-BFM 04, m-BACOD, DA-EPOCH-R, GPOH-HD 95, and ABVD. Regimens were selected for each case according to institutional preference and in accordance with the referred period. All patients who started their respective chemotherapy protocols had complete adherence during the period they were undergoing treatment. No patients were submitted to radiation or were elected for hematopoietic stem cell transplantation. Surgery was performed on some patients depending on each case.

Dose modifications and administration of granulocyte colony-stimulating factor (G-CSF) were used on selected patients at the discretion of the treating clinicians. According to national policy at the time, prophylaxis for opportunistic infections was given to all patients with AIDS-defining CD4+ T-cell counts. Finally, some patients received intravenous immunoglobulin at the decision of each center. Patients were screened for tuberculosis if there was clinical suspicion and treated if infected [32].

The outcomes for lymphoma treatment were complete response, progression of disease/non-complete response, relapse, or death.

### 2.7. Statistical Analysis

We used the measures of central tendency—i.e., means (standard deviation) or medians (interquartile range and range)—depending on the variable distribution type (normal or not) for numerical continuous variables. The Kolmogorov–Smirnov test and the Shapiro–Wilk tests were applied to verify the data normality assumption. Student’s t-test was used to compare the groups’ means. The Mann–Whitney test (for two-group comparisons) and Kruskal–Wallis test (for comparisons of three or more groups) were used for variables that did not meet the requirements for normal distribution. For categorical variables, the difference for each group’s percentage or risk was estimated using Pearson’s chi-square test or Fisher’s exact test.

Complete response was defined as: (i) complete disappearance of all target lesions and all nodes with long axes < 10 mm or (ii) 30% decrease in the sum of the longest diameters of the target lesions with normalization of the imaging exam. Progression of disease (non-complete response) was defined as: (i) >20% increase in the sum of the longest diameters of the target lesions; (ii) for small lymph nodes measuring < 15 mm post-therapy, a minimum absolute increase of 5 mm and a long diameter exceeding 15 mm; or (iii) appearance of a new lesion. Relapse was defined only for individuals who achieved complete response as the appearance of a new neoplastic lesion related to the original lymphoma [35,36,37,38]. We did not have any patients who achieved minor response or partial response in this study.

For survival analysis, the time of observation started on the date of lymphoma diagnosis (dynamic cohort time zero) and ended at the occurrence of the event or at the time of censoring. The outcomes of interest were (1) death by any cause for overall survival (OS), (2) an event (death by any cause or progression of disease/non-complete response or relapse, whichever came first) for event-free survival (EFS), and (3) relapse or death, whichever came first, in patients who achieved complete response for disease-free survival (DFS).

The 5-year OS was defined from the date of lymphoma diagnosis until the date of death in the first 5 years of follow-up. The 5-year EFS was calculated from the date of lymphoma diagnosis until the event in the first 5 years of follow-up. The 5-year DFS was calculated from the date of lymphoma diagnosis until the date of relapse or death in the first 5 years of follow-up for the patients who achieved complete response. The median follow-up time was calculated using the Kaplan–Meier (KM) curve. We did not have treatment abandonment or loss to follow-up in this study. All causes of death were also reviewed by two qualified study researchers (N.L.D. and M.G.P.L.).

Cox proportional hazard models were used to calculate the univariate hazard ratio (HR) for variables related to the risk and prognostic factor for the different types of survival calculated. Cox assumptions of proportionality were evaluated both graphically and analytically using a test based on Schoenfeld residuals. The variables with a confidence level of at least 80% (*p*-value < 0.20) were selected as possible candidates for multivariate analysis. After the multicollinearity assessment of every possible candidate variable, the most parsimonious model was built using the backward elimination method and the likelihood ratio test as selection criteria.

The R Project “coxphf” software package was used in cases of non-convergence of the likelihood function. This procedure implements Firth’s penalized maximum likelihood bias reduction approach for Cox regression, providing a solution in monotone likelihood cases (non-convergence of likelihood function) [39].

The cumulative incidence function of the different outcomes (death due to progression of disease/non-complete response, treatment-related death, or relapse) was calculated in the context of competing risks. We used the R Project “cmprsk” package, which implements competing risk analysis.

We also used the multiple imputation method (in SPSS Statistics software) to estimate the values of some relevant variables, such as CD4+ T-cell count and HIV viral load at lymphoma diagnosis. The missing values in these variables could have resulted from at least three possible causes: (1) testing was not available in the period referred to, (2) many patients died in a few days after hospital admission, and, finally, (3) in the early-treatment eras, the evaluation of this information was not recommended in national guidelines. The file with the imputed information was used only for the Cox multivariate analysis to verify if the results of the assessments performed with the original dataset (with missing values) would be corroborated. In this study, the results of the Cox multivariate analysis with the imputed data file are presented together with the conventional analysis.

Patient data were entered into a database (Microsoft Excel 2022) managed by a single researcher (N.L.D.). Lastly, all data processing and statistical analyses were performed using R Project version 4.0.2 and IBM SPSS Statistics software version 21.0.

## 3. Results

### 3.1. Characteristics of the Study Population

Between 1 January 1995, and 1 January 2018, 25 patients developed lymphomas among a total of 1306 patients with pediatric HIV infected via VT. Among these 25 children, 14 (56%) were male, and 11 (44%) were female. Further, 84% (21/25) had non-Hodgkin lymphomas (NHLs) and 16% (4/25) Hodgkin lymphomas (HLs). In the NHL group, 61.90% (13/21) had BLs, 28.57% (6/21) DLBCLs, 4.76% (1/21) ALCL CD30+/ALK+, and 4.76% (1/21) PTCL. Taken together, 90.47% (19/21) of NHLs had B-cell origins. All HL cases (4/4) were NSCHLs. No patients experienced more than one kind of cancer.

Sixty-four percent (16/25) of the lymphoma cases had diffuse involvement at presentation. In 16% (4/25), we only had lymph node involvement. Bone involvement occurred in 12% (3/25) of patients; in one case (4%), we had only gastrointestinal involvement, and in another case (4%), we had only bone marrow involvement. Only one patient had respiratory involvement (DLBCL), and only one patient had central nervous system (CNS) involvement (BL), which was not primarily nodal.

Among B-cell NHL patients, we found eight cases with diffuse abdominal involvement, five with small intestine involvement, five with bone (jaw, facial bones, and femur) involvement, and four with mediastinum involvement. Furthermore, there were four cases involving the liver, four involving bone marrow (BM), two involving the gallbladder, two involving the kidney, two involving the ovaries, one involving the CNS, one involving the soft palate, one involving the parotid, one involving the axillary lymph nodes, one involving the lungs, and one involving stomach infiltration. Fourteen cases (11 BLs and 3 DLBCLs) had more than one site of involvement and were considered as diffuse ones. Among the T-cell NHL cases, one (ALCL CD30+/ALK+) had submandibular lymph node and diffuse abdominal involvement (spleen, small intestine, and mesenteric lymph nodes). The other (PTCL) had mediastinum and BM infiltration. Lastly, we had four HL cases: one involved the cervical and thoracic lymph nodes, one involved the mesenteric lymph nodes, one involved BM, and one involved diffuse lymph nodes’ infiltration (cervical, thoracic, mesenteric, and inguinal).

The median age at lymphoma diagnosis was 7.43 years (range: 1.44–15.69; IQR: 4.55). B-cell NHLs were diagnosed at a median age of 7.11 years (range: 1.44–15.69 years; IQR: 4.98). The median age at HL diagnosis was 8.82 years (range: 5.89–11.37 years; IQR: 3.94). There were only two T-cell NHL cases: one diagnosed at 7.62 years (ALCL CD30+/ALK+), one at 8.20 years (PTCL).

In this study, only two children were exposed to prenatal antiretroviral therapy (AZT + DDI), starting at 32 weeks and 12 weeks of pregnancy in each case. These were also the only patients to receive antiretroviral therapy peripartum for 28 days after birth (AZT), respectively. Twenty children were treated with ART and seventeen with cART during the study period. Regarding the cART eras, we had 13 (52%) patients classified in the Early-cART era, 11 (44%) patients in the Mid-cART era, and one (4%) patient in the Late-cART era. Thirteen (52%) patients were receiving cART at the time of lymphoma diagnosis.

The median CD4+ T-cell count at HIV diagnosis was 16.00% (range: 2.00–34.00%; IQR: 17.55%). HIV infection was fully symptomatic with overt immunodeficiency (CDC subcategory C3) in nine of the 25 patients at HIV diagnosis. A total of five of the 25 cases, all in the C category, could not be assessed for CD4+ T-cell count categories, as patients had not had CD4+ counts performed in the eligible period for the study (three months before/three months after lymphoma diagnosis). The median HIV load at HIV diagnosis was 330,000 copies/mL (range: 5200–1,600,000; IQR: 793,000)**.**

The median CD4+ T-cell count at lymphoma diagnosis was 15.50% (range: 2.00–36.00%; IQR: 15.00%). In most patients (16 of 25), HIV infection was fully symptomatic with overt immunodeficiency (CDC subcategory C3) at lymphoma diagnosis.

Five children without severe immunosuppression (CDC stage one or two) and eleven with severe immunosuppression (CDC stage three) presented B-cell NHLs. Three of the HL cases were diagnosed with severe immunosuppression (CDC stage three). The two T-cell NHL cases were classified in CDC stage three.

Three patients did not have CD4+ T-cell counts performed during the whole period of the study, so they could not be evaluated in this analysis. Lastly, the median HIV load at lymphoma diagnosis was 78,000 copies/mL (range: 0–600,000; IQR: 274,000).

Eleven (44%) patients had PS 4 at lymphoma diagnosis. Fifteen (60%) patients had stage IV, IVA, or IVB at lymphoma diagnosis. Fourteen (56%) patients presented B symptoms. Among the 25 patients, three were not assessed for CNS involvement and two were not accessible for BM infiltration, as described in Table 1.

### 3.2. Lymphoma Treatment and Causes of Death

All 25 patients received appropriate chemotherapy for their cancer. The median number of days of chemotherapy was 129.00 (range: 1.00–626.00; IQR: 167.00). Most patients (16; 64%) were treated with NHL-BFM 95. One (4%) patient was treated with NHL-BFM 90, one (4%) with GPOH-HD 95, one (4%) with B-NHL-BFM 04, one (4%) with m-BACOD (a BL case), one (4%) with DA-EPOCH-R (a DLBCL case), and two (8%) with ABVD (both NSCHL cases). Two (8%) patients were treated with m-BACOD + NHL-BFM 95 (both BL cases). Two (8%) patients were submitted to surgery (bilateral oophorectomy, BL; small bowel resection, DLBCL). Lastly, all 13 of the patients who were in cART since lymphoma diagnosis were using cART at the beginning of the chemotherapy treatment period.

Eleven (44%) patients completed all prescribed chemotherapy and fourteen (56%) died before treatment completion.

Fifteen (60%) patients achieved complete response. Among them, three (20%) patients relapsed. In all, 18 (72%) patients died in this study.

Regarding the causes of death, ten (55.56%) patients were classified as progression of disease/non-complete response, three (16.67%) died after relapse, and five (27.78%) patients died even after achieving complete response as a result of chemotherapy toxicity (treatment-related death). All 25 patients had full adherence to lymphoma treatment.

The main population features and the outcomes of the 25 patients are summarized in Table 1.

### 3.3. Overall Survival, Event-Free Survival, and Disease-Free Survival for the Cohort

The median follow-up time for the cohort was 12.41 years (95% CI = 1.31–23.52 years). The 5-year OS probability for the entire cohort was 32.00% (95% CI = 13.72–50.23%), with a median survival of 0.88 years (95% CI = 0.04–1.71). Considering the PS, the 5-year OS for PS 4 was 0.00%, while for PS 1 to 3, it was 50.00% (95% CI = 24.00–76.00%).

Regarding EFS, the 5-year probability was 32.00% (95% CI = 13.72–50.23%), with a median survival of 0.88 years (95% CI = 0.04–1.71).

The 5-year DFS probability was 53.30% (95% CI = 28.02–78.58%), with a mean survival of 9.97 years (95% CI = 5.37–14.57).

Figure 1 shows the KM 5-year OS, EFS, and DFS probabilities for the cohort.

The HRs for the Cox proportional hazards models (uni- and multivariate analyses and multivariate analysis of imputed dataset) for death (OS) and event (EFS and DFS) occurrences are shown in Table 2, Table 3, and Table 4, respectively.

### 3.4. Prognostic Features

In both multivariate Cox regression analyses for the OS calculation, the risk of death was only related to PS 4 (HR: 4.85, 95% CI = 1.81–12.97, *p* = 0.002). In the EFS analysis, the prognostic factors were almost identical to the OS analysis (HR: 4.95, 95% CI = 1.84–13.34, *p* = 0.002), as the time between relapse and death for the three patients in question ranged from 0 to 30 days. Therefore, the time to event in each analysis was almost the same. The same values were found in the Cox multivariate analysis of the imputed dataset.

In the DFS analysis, only the CD4+ T-cell counts were considered a prognostic factor in the Cox multivariate analysis (HR: 0.86, 95% CI = 0.76–0.97, *p* = 0.017) and the Cox multivariate analysis of the imputed dataset (HR: 0.85, 95% CI = 0.75–0.95, *p* = 0.007).

All the assessments mentioned above are elucidated in Table 2, Table 3 and Table 4, respectively.

### 3.5. Competing Risk of Death due to Different Outcomes

The competing risks of different outcomes (death due to progression of disease/non-complete response, treatment-related death, or relapse) in 20 years were calculated (Figure 2). For death due to progression of disease/non-complete response, the competing risk was 40.00% (95% CI = 20.20–59.80%); for treatment-related death, it was 20.00% (95% CI = 3.65–36.35%); and for relapse, it was 12.57% (95% CI = 0.00–26.70%).

## 4. Discussion

This study aimed to establish survival probabilities associated with different outcomes and prognostic factors in CLWH infected by VT who developed lymphomas in Rio de Janeiro, Brazil, from 1 January 1995 to 1 January 2018. To the best of our knowledge, this is the first study evaluating this scenario in Brazil. Our cohort was characterized by advanced stage, poor PS, presence of B symptoms, high HIV viral load, and severe CDC category/stage at lymphoma diagnosis. The main features of the cohort were poor OS, EFS, and DFS probabilities with major difficulties in conducting standardized chemotherapy treatment, especially in the first months after the lymphoma diagnosis, despite cART coverage for more than half of the study patients.

Developing a trustworthy cancer study in an HIV-infected pediatric population is a challenging task (unfulfilled cancer case records, absence of hospital and population cancer registries in Rio de Janeiro, among other factors). Furthermore, the record of adherence to ART/cART and/or chemotherapy may not be accurate enough, especially in relation to antiretroviral medications. In addition, there are a variety of combinations of antiretroviral therapies and chemotherapy protocols, depending on the period and the pediatric oncohematologist’s choice among the different standard protocols usually employed at centers.

Brazil is thought to be a model of public response to the AIDS epidemic and has offered free access to ART and medications for opportunistic diseases through the public health system since the 1990s [28,40]. As described in a prior work by our group, in a portion of the Brazilian CLWH population, overall rates of ADMs decreased since the establishment of cART through the eras (Early-, Mid- and Late-cART eras) [23].

In our cohort, most of the lymphoma cases were male individuals, as presented in the literature [16,31]. The most frequent lymphoma group was NHL. B-cell NHL is related to HIV immunosuppression and includes all aggressive B-cell NHLs, such as BLs and DLBCLs [3,12]. In line with this, BLs and DLBCLs were the most common subtypes diagnosed in our study. This lymphoma profile has already been fully acknowledged by previous pediatric studies [41,42].

Furthermore, two different subtypes of T-cell lymphomas were found in our cohort: one ALCL CD30+/ALK+ and one PTCL. T-cell lymphomas are not common in HIV-positive individuals and account for only about 3% of lymphomas in people living with HIV (PLWH). They are usually diffuse and exhibit bone marrow involvement, as also seen in our study. As verified by Arzoo et al., patients with HIV-positive T-cell lymphomas have comparable prognoses and outcomes to B-cell HIV-related patients [19,43].

Lastly, the most recurrent HL subtype in PLWH is NSCHL, which is frequently associated with EBV infection [5]. All HL cases in our population were classified as the NSCHL subtype. This result is coherent with the EBV epidemiology of non-Caucasian ethnic groups living in developing and low-income regions [44].

In accordance with the literature [45,46], lymphomas observed in our cohort were mainly diffuse and demonstrated extranodal involvement and bulky disease. They were diagnosed at an early age. B-cell NHLs appeared at a median age of 7.11 years. BL was the only subtype of ADM that occurred in patients between 0 and 5 years of age. HLs were diagnosed at a median age of 8.82 years and T-cell NHLs in patients aged 7.62 (ALCL CD30+/ALK+) and 8.20 (PTCL).

As expected, most of the patients who developed lymphoma were from the early-cART era; in other words, they were admitted to the participating institution between 1995 and 1999, before cART implementation and its widespread use. Although most of our children were using ART/cART at lymphoma diagnosis, they did not start treatment with cART as the first antiretroviral therapy but with suboptimal treatments, which could explain the low cohort CD4+ T-cell counts at lymphoma diagnosis. Another important point to highlight is that we did not evaluate and describe viral suppression in this study. Viral load at HIV diagnosis was high in most patients, although it declined after the beginning of cART. Nonetheless, viral suppression was not achieved in most of the patients, even though nearly all supposedly adhered to the therapy. We were not able to attribute this to putative resistance drug since the recommendation in Brazilian Ministry of Health guidelines in the initial cART eras was to change antiretroviral therapy if there was a significant clinical worsening or CD4+ count deterioration, without regard for viral load counts. Furthermore, adequate drug resistance testing was not always available at the institutions in that period. Therefore, changing to a second-line cART regimen was not always performed, as would be recommended nowadays in the case of a positive result [9].

Starting cART as early as possible can help to avoid severe CD4+ lymphocyte suppression, restores immunological surveillance, and, furthermore, can decrease the risk of ADMs in CLWH, as well as the risk of death [20,21,23,46,47]. It is known that HIV pathogenesis varies in perinatally HIV-infected (perHIV) patients—a feature of our study population—due to age-related discrepancies in the immune system at the time of infection, route of transmission, and cART start time [45,48]. The reason for this is that the immune system of newborns is still immature and incapable of setting up an effective immune response. This results in high levels of HIV viremia, which, in turn, decrease with the introduction of cART [49]. As such, CLWH with vertically acquired infection showed a higher risk of ADMs, including some lymphoma subtypes, and thus malignancy can show faster progression in this context [23,50]. As one of the supposed effects of immune transformation promoted by cART, outcomes of HIV-infected cancer patients are known to be better when this therapy is used [23,24,51,52,53].

In our cohort, HIV diagnosis revealed fully symptomatic infection with the presence of immunodeficiency (CDC subcategories C2/3) in a high proportion of patients; that is, the patients generally arrived at the study institutions in precarious health conditions to start HIV treatment, which made it more difficult to revert the immunodeficiency state caused by the delay at the beginning of accurate antiretroviral treatment.

A significant portion of patients in our study were also not using cART at the time of the lymphoma diagnosis, most probably because of the lack of consensus regarding antiretroviral treatment scheduling in the early eras of HIV treatment. The first lymphoma diagnosis in our cohort was from 1996. At present, it is well-established that all children and adolescents should start cART at the time of HIV diagnosis, regardless of viral load or CD4+ T-cell counts [54]. Another reason for these children and adolescents being without cART use at lymphoma diagnosis could be that HIV and lymphoma identifications were concomitant, which occurred with four patients in this research (three BLs and one DLBCL).

As expected, HIV infection was fully symptomatic with evident immunodeficiency (CDC subcategory C3) in most CLWH with lymphomas. B-cell NHL essentially occurs in children and adolescents with severe immunosuppression. Although it is known that HL is not associated with HIV-induced immunosuppression [12,19], 75% of patients in our cohort who developed HL had severe immunodepression. Finally, the two T-cell NHL patients also had severe immunosuppression. The relationship between progressive CD4+ function suppression and cancer development is coherent with cART benefits in the prevention of opportunistic cancers through the restoration of the adaptive immune response [5].

In our study, PS 4 status at lymphoma diagnosis was an independent predictor of death risk; furthermore, poor performance status was observed in several CLWH at diagnosis, frequently associated with advanced lymphoma. Moreover, the results of the EFS analysis were very similar to those of the OS analysis. Although the event occurrence times in the initial parts of the OS and EFS curves were different, the 5-year survival probabilities converged to 32.00% with the same standard error (SE) and, consequently, the same 95% CI. This was due to the fact that, for the three disease relapses in the study, the “Death” event occurred very shortly after. Continuing in this analysis, for many patients, symptoms related to lymphoma led to them seeking medical care, and thus HIV was detected concomitantly. This possible delay in detection can be attributed to family, caregiver, and health system factors, such as low levels of family education, healthcare access difficulties, and insufficient qualifications among primary healthcare professionals for early cancer diagnosis [17,20,55]. Early recognition of malignancies in CLWH is a challenge due to overlapping clinical features [27]. As such, our observations are in line with WHO recommendations about the importance of early HIV screening and diagnosis for all children and adolescents for better prognosis [18,56].

Linked to PS, most patients presented stage IV disease (which includes IV, IVA, and IVB) at lymphoma diagnosis. This was expected since lymphomas related to HIV are generally aggressive and usually are presented at a late stage at diagnosis [12,13,19,57]. As opposed to previous studies, the late stage did not predict poor survival outcomes in our cohort, and we can probably attribute this to our sample size and the high collinearity between PS status and staging in this scenario [14,15,31].

Additionally, and in line with prior works [9,20,31], CD4+ T-cell counts—as a continuous variable—were the only predictor in the DFS analysis in this survey; i.e., low CD4+ counts worsened the prognosis. In the DFS univariate analysis, PS 4 status was found to be not significant for the prediction of relapse or death in the complete response group. It is possible that these patients had a lower PS at lymphoma diagnosis and, as a result, better withstood the usually intensive chemotherapy regimen at the beginning of treatment. It can be speculated that PS 4 is the best predictor of early death and low CD4+ T-cell counts are the best predictor of late death in complete response patients.

It is important to mention that other factors, such as age, biological sex, B symptoms, and histological subtype, among others, did not achieve statistical significance, probably due to our small sample. Lists of such prognosis factors have even been presented in other studies [9,14,19,21]. As these studies usually have small samples and, consequently, low statistical power, it is common to find only one of these prognostic factors as predictors of survival, and not a combination of them, in the final multivariate model. Furthermore, we did not evaluate events such as anemia, neutropenia, or thrombocytopenia; erythrocyte sedimentation rates (ESRs); or lactate dehydrogenase (LDH) measures, among other laboratory parameters. It is an arduous task to find reliable and continuous results in medical charts for a 23 year study evolution without a central laboratory to perform the analysis and with so many different reference values in question.

Concerning lymphoma treatment, the fact that most patients were using cART during the cancer protocols and the number of days of chemotherapy utilization were not prognostic factors in the different survival analyses. The same occurred for the chemotherapy regimens applied [14,58]. Another important point is that more than half of the patients did not complete the chemotherapy regimens. This was not caused by abandonment but by early death. Of the 25 patients, ten were classified as progression of disease/non-complete response and, from the remaining 15, three relapsed.

The competing risk analysis showed the main outcomes that compromised the survival of our studied group. It is impressive that relapse was the least important one. As suggested by Howard et al. [59], we need strategies that reduce the deaths that occur before complete response can be achieved, as well as the treatment-related deaths among patients in continuous complete remission. An important caveat is that infection was a significant cause of death in our cohort. These conditions occurred mainly after treatment had started, indicating that these patients would not have tolerated chemotherapy very well, which was seen in five cases who achieved complete response. A multidisciplinary team comprising oncohematologists, HIV clinicians, and infectious diseases specialists is desirable to ensure that both expected infections associated with neutropenia and HIV-specific opportunistic infections are adequately managed [9,16,18,27,59].

The strength of our study was the fact that it used a multicentric cohort with high-quality pathologic analyses, encompassing a long calendar period with a robust follow-up median time. Furthermore, we had complete retention and outcome ascertainment. The limitations of this work resulted from the fact that it was a hospital-based study, as there is no population-based register for cancer in Rio de Janeiro. Additionally, it was a retrospective study with a 23 year follow-up and, consequently, medical records were not fully available. Access to lymphoma biopsies was challenging and the quality of the original material tended to be a problem. Reclassification of all tumor samples in accordance with WHO guidelines was sometimes impossible due to the loss or deterioration of some tumor samples. This occurred in five cases and, therefore, we had to assume that the reports given by competent pathologists at the time of lymphoma diagnosis were correct. Furthermore, we did not evaluate some important laboratory parameters, and there was a lack of CNS/BM evaluation, as well as CD4+ T-cell counts, for some patients. Lastly, the evaluation of the degree of adherence to cART and/or chemotherapy was a challenge, as previously mentioned. We did not find other similar studies that analyzed the survival probabilities and prognostic factors in a similar cohort in Brazil to use as comparisons. This paper therefore presents important information about the Brazilian scenario.

## 5. Conclusions

This multicentric study was based on a group of CLWH assisted at five recognized centers for HIV and cancer care in Rio de Janeiro, Brazil. It illustrates for the first time the survival probabilities and prognostic factors in children and adolescents vertically infected by HIV who developed lymphomas. In summary, the lower survival probabilities in our cohort appear to have been associated with poor PS and low CD4+ T-cell counts. Unfortunately, the small sample size did not allow a more extensive analysis. In addition, we observed that, in our study population, most lymphomas were diagnosed in an overt immunodeficiency state and at a late stage of the disease. In some cases, lymphomas were the first HIV-presenting signs. In this context, starting cART as soon as possible, as well as maintaining satisfactory adherence, is essential for better prognosis.

Regardless of the findings presented in this paper, it is still unknown if lymphoma survival and prognosis among CLWH in Brazil are similar to those in high-income countries. This work was designed to find ways to improve the quality of HIV-related lymphoma treatment in our population. For the future, we must evaluate how the knowledge about prognostic factors obtained by this study in relation to overall survival (PS 4) can be reflected in the supportive treatment and choice of therapeutic protocols for patients with greater frailty. We believe that patients with very advanced PS could benefit from low-intensity protocols until their general clinical condition improves, and only then will they be able to reach the intensity levels of the most effective protocols for their baseline lymphoma. Regarding disease-free survival, low T-cell CD4+ counts should alert pediatricians of the need to verify adherence to cART for the rest of patients’ lives after the diagnosis of lymphoma and to adopt supportive measures for possible concurrent immunodeficiency states, such as prolonged neutropenia (use of G-CSF, preemptive diagnosis of fungal infections, and prophylaxis for opportunistic infections) and low immunoglobulin counts (immunoglobulin replacement if applicable). For more definitive results, a larger number of patients would have to be enrolled in a wider multicenter study to accurately determine further predictors for poor outcomes. Additionally, in the case of a prospective study, which would probably be on a global scale because of the rarity of HIV-related lymphomas in the post-cART era, standard HIV and lymphoma treatments must be employed in order to obtain valid and accurate results.

In addition to greater adherence to cART by the pediatric population living with HIV, the number of children and adolescents with HIV-related lymphoma should decrease further with the reduction in VT. To this end, it is essential to ensure broad access to VT prevention strategies locally and globally. This would include offering HIV testing to all pregnant women during prenatal care; disseminating knowledge about the risk factors associated with HIV transmissibility; and use of ART during pregnancy, postpartum, and for newborns if applicable, in addition to other strategies of combined prevention.

## Figures and Tables

**Figure 1 cancers-15-02292-f001:**
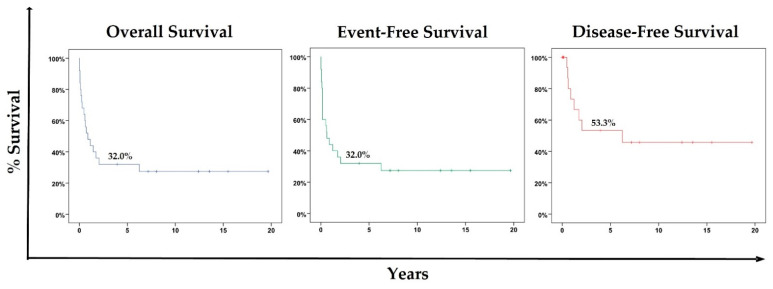
Five-year Kaplan–Meier (5-year KM) OS, EFS, and DFS probabilities for the cohort.

**Figure 2 cancers-15-02292-f002:**
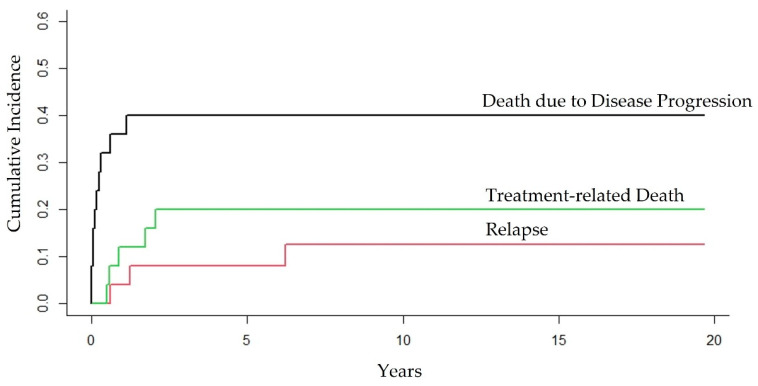
Cumulative incidence (competing risk model) for death due to disease progression/non-complete response, treatment-related death, and relapse over 20 years of evolution.

**Table 1 cancers-15-02292-t001:** Population features.

Features	*n* = 25 (100%)
Biological Sex	
Female	11 (44%)
Male	14 (56%)
Lymphoma Subtype	
BL (ADM)	13 (52%)
DLBCL (ADM)	6 (24%)
NSCHL (NADM)	4 (16%)
ALCL (CD30+/ALK+) (NADM)	1 (4%)
PTCL (NADM)	1 (4%)
Site of Lymphoma at Disease Presentation	
Lymph node	4 (16%)
Respiratory	0
Gastrointestinal	1 (4%)
Bones	3 (12%)
CNS	0
BM	1 (4%)
Diffuse (two or more different sites)	16 (64%)
Year of Lymphoma Diagnosis	
1996	1 (4%)
1998	2 (8%)
1999	2 (8%)
2000	3 (12%)
2001	2 (8%)
2002	4 (16%)
2004	4 (16%)
2005	3 (12%)
2007	1 (4%)
2010	2 (8%)
2013	1 (4%)
Age at Lymphoma Diagnosis (years)	
Median (IQR)	7.43 (4.55)
0–4	5 (20%)
5–9	13 (52%)
≥10	7 (28%)
ART Prophylaxis	
Yes	02 (8%)
No	23 (92%)
ART Use?	
Yes	20 (80%)
No	5 (20%)
cART Use?	
Yes	17 (68%)
No	8 (32%)
cART Era	
Early-cART	13 (52%)
Mid-cART	11 (44%)
Late-cART	1 (4%)
cART at Lymphoma Diagnosis?	
Yes	13 (52%)
No	12 (48%)
CD4+ T-cell Count at HIV Diagnosis (%)	
Median (IQR)	16.00% (17.55%)
>25%	4 (20%)
15–25%	7 (35%)
<15%	9 (45%)
NP	5
HIV Load at HIV Diagnosis (Copies/mL)	
Median (IQR)	330,000 (793,000)
CDC Category/Stage at HIV Diagnosis	
N2	1 (4%)
A1	2 (8%)
A2	4 (16%)
C	5 (20%)
C1	2 (8%)
C2	2 (8%)
C3	9 (36%)
CD4+ T-cell Count at Lymphoma Diagnosis (%)	
Median (IQR)	15.50 % (15.00%)
>25%	2 (11.10%)
15–25%	8 (44.40%)
<15%	8 (44.40%)
NP	07
HIV Load at Lymphoma Diagnosis (Copies/mL)	
Median (IQR)	78,000 (274,000)
CDC Category/Stage at Lymphoma Diagnosis	
C	3 (12%)
C1	1 (04%)
C2	5 (20%)
C3	16 (64%)
Performance Status (PS)	
1	5 (20%)
2	9 (36%)
3	0
4	11 (44%)
Stage of Lymphoma	
I	1 (4%)
IA	1 (4%)
III	2 (8%)
IIIB	6 (24%)
IV	6 (24%)
IVA	1 (4%)
IVB	8 (32%)
B Symptoms	
Yes	14 (56%)
No	11 (44%)
CNS Involvement	
Yes	1 (4.54%)
No	21 (95.45%)
NP	3
BM Involvement	
Yes	6 (26.08%)
No	17 (73.91%)
NP	2 (8%)
Chemotherapy Protocols	
NHL-BFM 90	1 (4%)
GPOH-HD-95	1 (4%)
NHL-BFM 95	16 (64%)
B-NHL-BFM 04	1 (4%)
m-BACOD	1 (4%)
DA-EPOCH-R	1 (4%)
ABVD	2 (8%)
NHL-BFM 95 + m-BACOD	2 (8%)
Days of Chemotherapy	
Median (IQR)	129.00 (167.00)
Chemotherapy Completion	
Yes	11 (44%)
No (death)	14 (56%)
cART at Chemotherapy?	
Yes	13 (52%)
No	12 (48%)
Surgery	
Yes	2 (8%)
No	23 (92%)
Outcomes	*n* = 25 (100%)
Complete Response	
Yes	15 (60%)
No	10 (40%)
Relapse	
Yes	3 (20%)
No	12 (80%)
Death	
Yes	18 (72%)
No	7 (28%)
Status at Moment of Death	
Progression of disease/non-complete response	10 (55.56%)
After relapse	3 (16.67%)
Complete response	5 (27.78%)

Abbreviations: AIDS-defining malignancy (ADM); anaplastic large cell lymphoma (ALCL CD30+/ALK+); bone marrow (BM); Burkitt lymphoma (BL); combined antiretroviral therapy (cART); central nervous system (CNS); diffuse large B-cell lymphoma (DLBCL); interquartile range (IQR); nodular sclerosis classical Hodgkin lymphoma (NSCHL); non-AIDS-defining malignancy (NADM); not applicable (NA); not performed (NP); peripheral T-cell lymphoma (PTCL).

**Table 2 cancers-15-02292-t002:** Hazard ratio (HR) for death (OS) (Cox proportional hazards models).

	Univariate				Multivariate				Multivariate Analysis of Imputed Dataset			
Variables (*n*)	HR	Low	High	*p*	HR	Low	High	*p*	HR	Low	High	*p*
Biological Sex												
Female (11)	0.88	0.55	1.41	0.547								
Male (14)	1											
Year of Lymphoma Diagnosis (25)	0.98	0.88	1.09	0.760								
Age at Lymphoma Diagnosis (25)	0.99	0.87	1.15	0.976								
Lymphoma Group												
ADM (19)	0.62	0.22	1.79	0.380								
NADM (6)	1											
cART Era												
Early-cART (13)	0.24	0.03	2.15	0.204								
Mid-cART (11)	0.34	0.04	2.95	0.326								
Late-cART (1)	1											
CD4+ T-Cell Count at Lymphoma Diagnosis (%) (18)	0.91	0.84	0.98	0.014								
NP (7)												
HIV Load at Lymphoma Diagnosis (Copies/mL) (17)	1.00	1.00	1.00	0.797								
PS at Lymphoma Diagnosis												
4 (11)	4.85	1.82	12.97	0.002	4.85	1.82	12.97	0.002	4.85	1.82	12.97	0.002
<4 (14)	1											
Lymphoma Stage												
I–III (10)	1											
IV (15)	1.53	0.57	4.09	0.398								
cART at Chemotherapy												
Yes (13)	0.50	0.19	1.26	0.142								
No (12)	1											

Abbreviations: AIDS-defining malignancy (ADM); combined antiretroviral therapy (cART); hazard ratio (HR); non-AIDS-defining malignancy (NADM); not performed (NP); performance status (PS).

**Table 3 cancers-15-02292-t003:** Hazard ratio (HR) for events (EFS) (Cox proportional hazards models).

	Univariate				Multivariate				Multivariate Analysis of Imputed Dataset			
Variables (*n*)	HR	Low	High	*p*	HR	Low	High	*p*	HR	Low	High	*p*
Biological Sex												
Female (11)	0.75	0.29	1.93	0.549								
Male (14)	1											
Year of Lymphoma Diagnosis (25)	0.99	0.89	1.10	0.798								
Age at Lymphoma Diagnosis (25)	1.00	0.87	1.16	0.978								
Lymphoma Group												
ADM (19)	0.58	0.20	1.65	0.306								
NADM (6)	1											
cART Era												
Early-cART (13)	0.37	0.04	3.16	0.367								
Mid-cART (11)	0.54	0.07	4.49	0.571								
Late-cART (1)	1											
CD4+ T-Cell Count at Lymphoma Diagnosis (%) (18)	0.90	0.84	0.98	0.012								
NP (7)												
HIV Load at Lymphoma Diagnosis (Copies/mL) (17)	1.00	1.00	1.00	0.777								
PS at Lymphoma Diagnosis												
4 (11)	4.95	1.84	13.34	0.002	4.95	1.84	13.34	0.002	4.95	1.84	13.34	0.002
<4 (14)	1											
Lymphoma Stage												
I–III (10)	1											
IV (15)	1.53	0.57	4.09	0.398								
cART at Chemotherapy												
Yes (13)	0.49	0.19	1.26	0.142								
No (12)	1											

Abbreviations: AIDS-defining malignancy (ADM); combined antiretroviral therapy (cART); hazard ratio (HR); non-AIDS-defining malignancy (NADM); not performed (NP); performance status (PS).

**Table 4 cancers-15-02292-t004:** Hazard ratio (HR) for events (DFS) (Cox proportional hazards models).

	Univariate				Multivariate				Multivariate Analysis of Imputed Dataset			
Variables (*n*)	HR	Low	High	*p*	HR	Low	High	*p*	HR	Low	High	*p*
Biological Sex												
Female (11)	0.68	0.16	2.89	0.607								
Male (14)	1											
Year of Lymphoma Diagnosis (25)	0.92	0.78	1.09	0.355								
Age at Lymphoma Diagnosis (25)	1.03	0.85	1.25	0.745								
Lymphoma Group												
ADM (19)	0.46	0.08	2.45	0.366								
NADM (6)	1											
cART Era *												
Early-cART (13)	0.73	0.19	2.87	0.645								
Mid-cART (11)	1.36	0.35	5.35	0.645								
Late-cART (1) **	NA	NA	NA	NA								
CD4+ T-Cell Count at Lymphoma Diagnosis (%) (18)	0.86	0.76	0.97	0.017	0.86	0.76	0.97	0.017	0.85	0.75	0.95	0.007
NP (7)												
HIV Load at Lymphoma Diagnosis (Copies/mL) (17)	1.00	1.00	1.00	0.942								
PS at Lymphoma Diagnosis												
4 (11)	3.45	0.82	14.60	0.092								
<4 (14)	1											
Lymphoma Stage												
I–III (10)	1											
IV (15)	2.25	0.45	11.28	0.323								
cART at Chemotherapy												
Yes (13)	0.88	0.21	3.72	0.867								
No (12)	1											

Abbreviations: AIDS-defining malignancy (ADM); combined antiretroviral therapy (cART); hazard ratio (HR); non-AIDS-defining malignancy (NADM); not performed (NP); performance status (PS). * For this analysis, dummy variables were used. ** The values did not converge even with the penalized function being applied.

## Data Availability

The data presented in this study are available on request from the corresponding author.

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
