# Peer review of "Prognostic Factors in Children and Adolescents with Lymphomas and Vertical Transmission of HIV in Rio de Janeiro, Brazil: A Multicentric Hospital-Based Survival Analysis Study"

_cancers, 2023, doi:10.3390/cancers15082292_

Round 1
Reviewer 1 Report
1)Although this paper is primarily about lymphomas in children infected with HIV, and not about HIV infection, I think there should be a bit of discussion about how highly active antiretroviral therapy has nearly eliminated pediatric HIV infection. The paper states "Only two patients had been exposed to prenatal antiretroviral therapy (AZT + DDI) starting at 32 weeks and 12 weeks, respectively, and only these children received antiretroviral therapy peripartum and for 28 days after birth (AZT)." and this is presumably 2 of the 25 who developed lymphoma and not two of the 1,306 cases of pediatric HIV infection. Does this mean that most of the 25 infections (or most of the 1,306 infections) were due to lack of treatment of the mothers during pregnancy?
2) In the abstract "The 5-year Overall Survival (OS) and 5-year Event-Free Survival (EFS) probabilities were 32.00% (95% CI = 13.72%–50.23%)." I was expecting 2 numbers for OS and EFS.
3) "The study period from 1995 to 2018 was divided into three subperiods"... But I don't see those subperiods broken out in the tables, except that 13 were pre-cART, 11 were mid-cART and one Late-cART.
4) Most of the lymphomas (19 of the 25) were AIDS-defining malignancies, and most (19) were of B-cell origin. I would have guessed that integration of HIV into T-cell (CD4+ cells) would have caused lymphomas, so I am surprised they are mostly B-cell. Is it B-cell activation that causes the lymphoma?
5) "The median CD4+ T-cell count at HIV diagnosis was 16.00%..." and "The median CD4+ T-cell count at lymphoma diagnosis was 15.50%..." I was expecting a number, like 200 cells/ml and not a percentage.
6) "This work was designed to find ways to improve the quality of lymphoma-related-to-HIV treatment in our population." But it seems that the major factor here is using antiretroviral therapy to prevent pediatric infection in the first place. And if that fails, it seems that good antiretroviral therapy prevents most lymphomas. Most of these patients did not get good therapy early, and were in AIDS decline when the lymphomas were diagnosed.
Reviewer 2 Report
I suggest the authors to make the following corrections:
1. on page 2, in the abstract, Row 4, after ")" must be deleted "." and word "and" must be written with a small letter;
2. in the Introduction, in the first paragraph, there are too many bibliographical references to the same sentence....."[1-9]"; likewise in paragraph 3, sentence 1;
3. on page 3, the title from 2.4 must be reformulated;
4. on page 4, Row 8, the word "sex" must be replaced by "gender";
5. on page 4, at 2.7, sentence 3, it must be written..."The t Student test";
6. on page 5, in Row 2, it must be written..."evaluated by Pearson Chi-square test";
7. on page 5, on Row 10, without "." before [36-39];
8. on page 7, the title from table 1 must be changed, the word "sex" must be replaced with "gender", the word "female" with "feminine" and the word "male" with "masculine";
9. the same corrections must be made in tables 2, 3 and 4;
10. on page 16, the word "cohort" must be replaced with "studied group" in order not to be confused with the cohort study (in medical research, cohort means the group of people born in the same year);
11. on page 18, in the conclusions, the first sentence should be reformulated, I suggest "This multicentric study is based on parents who".......
12. before the conclusions, a few sentences should be added regarding the limits of the study;
13. since the study group includes lymphoma patients with and without HIV, a more suitable title for the article must be found.
Congratulations for the work done!
